# Altered Grey Matter-Brain Healthcare Quotient: Interventions of Olfactory Training and Learning of Neuroplasticity

**DOI:** 10.3390/life13030667

**Published:** 2023-02-28

**Authors:** Keita Watanabe, Keisuke Kokubun, Yoshinori Yamakawa

**Affiliations:** 1Institution of Open Innovation, Kyoto University, Kyoto 606-8501, Japan; 2Smart-Aging Research Center, Tohoku University, Sendai 980-8575, Japan; 3Institute of Innovative Research, Tokyo Institute of Technology, Tokyo 152-8550, Japan; 4Academic and Industrial Innovation, Kobe University, Kobe 657-8501, Japan; 5ImPACT Program of Council for Science, Technology, and Innovation (Cabinet Office, Government of Japan), Tokyo 100-8914, Japan; 6BRAIN IMPACT General Incorporated Association, Kyoto 606-8501, Japan

**Keywords:** olfactory training, learning of neuroplasticity, curiosity, grey matter, brain healthcare quotient

## Abstract

Recent studies revealed that grey matter (GM) changes due to various training and learning experiences, using magnetic resonance imaging. In this study, we investigate the effect of psychological characteristics and attitudes toward training and learning on GM changes. Ninety participants were recruited and distributed into three groups: an olfactory training group that underwent 40 olfactory training sessions designed for odour classification tasks, a group classified for learning of neuroplasticity and brain healthcare using a TED Talk video and 28 daily brain healthcare messages, and a control group. Further, we assessed psychological characteristics, such as curiosity and personal growth initiatives. In the olfactory training group, we conducted a questionnaire survey on olfactory training regarding their interests and sense of accomplishment. In the olfactory training group, the GM change was significantly correlated with the sense of achievement and interest in training. The learning of neuroplasticity and brain healthcare group showed a significantly smaller 2-month GM decline than did the control group. The Curiosity and Exploration Inventory-II scores were significantly correlated with GM changes in both intervention groups only. In conclusion, our result suggested that training or learning with a sense of accomplishment, interest, and curiosity would lead to greater GM changes.

## 1. Introduction

The decline in grey matter (GM) volume is a highly visible aspect of the chronological ageing process, resulting from neural shrinkage and neuronal loss [1]. GM volume gradually decreases after 20 years of an individual’s life [2]. The age-related decline in GM volume is associated with cognitive decline [3,4]. Because a globally ageing society has become a social problem, elucidating methods to prevent the decline in GM volume, the so-called brain reserve, is an important topic [5].

Neuroplasticity and neural reorganisation due to learning and training can be considered one method of brain reserve. Contrary to assumptions that changes in the brain are possible only during critical periods of development, modern neuroimaging studies revealed that after adulthood, learning and training could alter GM in the brain area corresponding to the task [6,7]. After 3 months of juggling training, older individuals showed larger GM volumes in the visual cortex [8]. In addition, 3-month intensive learning of medical students [9] and 8-week intensive memory training of older participants [10] induced increased hippocampal volume and cortical thickness in the orbitofrontal and fusiform cortex, respectively. Although increased GM volume due to intensive training and learning can renormalise [11], protective effects against the decline of GM volume can be expected [12]. Interestingly, even if the training is the same, GM alternations have individual differences. In a sequential pinch force task study, participants with strong performance improvements had larger increases in GM volume. However, participants with small behavioural gains showed no change or even a decrease in GM volume [13]. One possible factor contributing to individual differences is the attitude towards training and learning or psychological characteristics, such as interest and curiosity. For instance, in the neurocognitive study, high interest in learning has been shown to have a positive impact [14]. However, to our knowledge, no studies have investigated the relationship between the amount of GM change and attitudes or psychological characteristics. Therefore, here, we investigated the effects of attitude and psychological characteristics on GM alterations. Training and learning could be especially effective for participants who enjoy training and take high self-growth initiatives.

Various studies revealed that many different types of training and learning, such as motor, language, memory, and sensory, can alter GM [6,15]; of these, olfactory training has been validated to alter GM in several studies. In addition, it has been suggested that olfactory training not only improves the sense of smell, but also may lead to improved quality of life [16,17]. GM alterations due to olfactory training have been suggested in patients with olfactory loss and healthy participants [18,19,20,21]. For example, in a 6-week controlled training that included odour intensity classification, odour quality classification, and target odour detection tasks, 12 healthy participants (aged 18–35 years) showed increased cortical thickness in the entorhinal cortex, fusiform gyrus, and inferior frontal gyrus [21]. Unlike other training and learnings (e.g., juggling and intensive learning of medical students), olfactory training directly uses the innate sense of smell.

A possible method to improve the brain reserve, other than directly recommending training or learning, would be to encourage behaviour change for new initiatives. For instance, diversity of daily activities is associated with greater hippocampal volume [22]. This result supports the importance of new initiatives for brain reserve. As a method to encourage behaviour change for new initiatives, we hypothesised that communicating knowledge of neuroplasticity and brain healthcare would be useful. A previous study reported that during an intervention of teaching about the depiction of growing neural pathways, the key message was that learning changes the brain and improves academic achievement in junior high school students [23]. In addition, previous functional magnetic resonance imaging (MRI) studies have found that health messages can lead to healthy behaviour through brain activation [24,25].

In this current study, we considered evaluating GM alterations due to training and learning as a whole brain. Interestingly, learning and training led to an increase and decrease in the GM volume in different brain regions. In a previous study involving medical students [9], decreased occipitoparietal lobe volume and increased parietal lobe and hippocampal volumes resulted from intensive learning. The measurement of total GM volume can be considered a method to evaluate GM alterations in the entire brain. However, the implications of a 1 mL GM volume change can differ between the hippocampus, with a total volume of 3 mL, and the occipitoparietal lobe. Therefore, to assess GM alterations using the interventions, we used the GM-brain healthcare quotient (BHQ) of the International Telecommunication Union standards (H. 861.1), representing GM volume as a standardised score in each anatomical brain region. In a previous study, GM-BHQ was more strongly correlated with cognitive function than hippocampus- or parahippocampus-volumetry [26].

In this study, we evaluated brain changes through two types of interventions: olfactory training using incense and learning about neuroplasticity and brain healthcare. We also investigated attitudes toward training and psychological characteristics. Our first aim was to clarify whether attitudes and psychological characteristics affect neuroplasticity due to training. Our second aim was to investigate whether the brain would become healthier by acquiring scientific knowledge regarding brain healthcare.

## 2. Materials and Methods

### 2.1. Participants

Human investigations were performed in accordance with the guidelines provided and approved by the Institutional Review Board (approval number A20152). All participants provided written informed consent to participate in this study.

A marketing agency recruited participants who met the inclusion criteria. The inclusion criteria were that the participants had (a) no problems with an MRI, such as body metal, tattoos, or pregnancy; (b) had no underlying disease, psychiatric disorders, brain disease, or lifestyle-related diseases; and (c) were not heavy smokers or drinkers, defined as an intake of 25 or more cigarettes or 20 g or more alcohol per day. Depression tendencies were considered an exclusion criterion if the Centre for Epidemiologic Studies Depression scale score [27] was over 16 when the participants applied. The marketing agency continued to recruit the participants until their number reached the requirements—aged 30s: men = 9, women = 9; 40s: men = 12, women = 12; 50s: men = 12, women = 12; and 60–64: men = 12, women = 12. A total of 90 participants were classified into three groups, with stratified randomisation for sex and age groups, namely, (1) control group, (2) olfactory training group, and (3) the learning of neuroplasticity and brain healthcare group. Figure 1 shows the flow diagram of included participants. In addition, a radiologist reviewed the MRI scans to exclude participants with cerebrovascular lesions or other gross MRI abnormalities. None of the participants were excluded because of MRI abnormalities.

The figure shows a flow chart of enrolment, randomisation, exclusion, and analysis. We evaluated 29 participants in the control group, 30 in the olfactory training group, and 30 in learning neuroplasticity and brain health care.

### 2.2. Intervention Design

All participants underwent a psychological characteristics test and their first MRI in March 2021. The psychological characteristics test was undertaken online within a week before the first MRI. The intervention procedures were then performed in the olfactory training and learning of neuroplasticity and brain healthcare groups (Figure 2). Approximately 2 months later, in May 2021, a second MRI examination was administered, and a training questionnaire was distributed.

The figure shows an experimental intervention design for two types of intervention: (a) olfactory training and (b) learning neuroplasticity and brain healthcare. This study assessed changes in the brain for 2 months due to the interventions. We referred to a previous study to create a figure [28].

### 2.3. Olfactory Training

The training content was explained to the participants on the first day of MRI acquisition. For approximately 2 months, 15 min of olfactory training at home using incense was conducted 5 days a week for a total of 40 times. The training was designed by an incense company and consisted of an odour classification task. In the first training session, the participants formed three pairs based on three types of odours, with six scent bags in total. Olfactory training was designed to increase the difficulty level gradually; 1st to 8th day: 3 odour types; 9th to 20th day: 6 odour types; 21st to 32nd days: 9 odour types; and 33rd to 40th days: 12 odour types. The types of odours were labelled on scent bags. Participants turned the labelled side backwards and shuffled the scent bags. Then, participants sorted the bags into combinations of types of odours. Participants scored the number of correct answers based on labels and recorded them on the web after each training session.

### 2.4. Learning Neuroplasticity and Brain Healthcare

The goals of learning neuroplasticity and brain healthcare were to know (a) whether a new initiative can change brain volumes and (b) how to keep the brain healthy. The first learning session was conducted at an MRI facility immediately after the first MRI acquisition. In the session on learning neuroplasticity at an experimental space, the participants watched a YouTube video of the TED Talk by Lara Boyd on the same topic [29]. In this video, neuroplasticity was explained in an easy-to-understand manner for non-scientists. Further, the video explains that new initiatives will change oneself, including the brain. Afterwards, a brief test of neuroplasticity was performed to confirm comprehension (Appendix A). All participants either got a perfect score or got one question wrong. A total of 28 brain healthcare messages (Appendix A) were emailed to the participants every morning on every weekday. To confirm that the participants read the message, they evaluated the helpfulness of each message using a 5-point scale on the web.

### 2.5. Psychological Characteristics

Psychological characteristics and perceived stress conditions were assessed on the web within 1 week before MRI acquisition. The assessed items were as follows: Personal Growth Initiative Scale-II) [30,31], Satisfaction with Life Scale [32], Subjective Happiness Scale [33,34], Curiosity and Exploration Inventory-II [35], Short Grit Scale (Grit-S) [36], Rosenberg Self-Esteem Scale [37,38], and Perceived Stress Scale [39] (Table 1 and Table 2).

Personal growth initiative, which is the active and intentional desire to grow as a person, has consistently shown positive relations with growth [40]. Curiosity, a motivational state that stimulates exploration, is assumed to fundamentally impact learning [41]. Grit, a learner’s competence to continue after complications, is associated with growth [42]. In addition, well-being can promote the learner’s success [43]. Well-being is related to subjective happiness, overall happiness based on one’s own perspective, and self-esteem, global feelings of self-liking and self-worth [44]. The Satisfaction with Life Scale was developed to measure subjective well-being [32]. Conversely, the Perceived Stress Scale measures individual stress levels. Severe stress may negatively affect training and learning [45]. In all these psychological characteristics tests, high scores indicate high psychological characteristics, such as high personal growth, initiative, and curiosity. In the Perceived Stress Scale, high scores indicate high stress levels.

### 2.6. Questionnaire for Olfactory Training

A questionnaire for the olfactory training group was administered after the second MRI acquisition to investigate attitudes toward training. The questionnaire included the factor of ‘enjoyment’, ‘sense of achievement’, ‘burden’, ‘fatigue’, and ‘interest’ regarding the training. In addition, subjective improvement in the sense of smell or taste was assessed. The questionnaire contents are presented in Table 3. Participants indicated how true each item was on a 5-point scale, with scores ranging from 1 (strongly disagree) to 5 (strongly agree).

### 2.7. Questionnaire for Behavioural Changes

Intervention through learning neuroplasticity and brain healthcare required that participants start a new initiative for brain healthcare. Therefore, we asked all participants of the three groups to use a questionnaire and determine whether they started a new initiative between the first and second MRI. The questionnaire content is described in Appendix A. Scores for all 10 items were summed, with higher scores representing more new initiatives.

### 2.8. MRI

MRI was performed using a Siemens 3.0 T MR system with a 32-channel head array coil. Original T1 images were acquired using magnetisation-prepared rapid gradient echo. The acquisition parameters were as follows: repetition time, 1,900 ms; echo time, 2.5 ms; inversion time, 900 ms; flip angle, 9; matrix size, 256 × 256; field of view, 256 mm; slice thickness, 1 mm; and voxel size, 1 × 1 × 1 mm.

### 2.9. Image Processing for Brain Volume

Structural imaging data were processed using SPM12 software program (Statistical Parametric Mapping 12; Institute of Neurology, London, UK) [46,47]. The three-dimensional-T1-weighted imaging in native space was spatially normalised, segmented into GM, white matter, and cerebrospinal fluid images. These images were modulated using the Diffeomorphic Anatomical Registration Through Exponential Lie Algebra toolbox in SPM12 [46,48]. The modulated GM images were smoothed using an 8-mm full-width at a half-maximum Gaussian kernel. Proportional GM images were generated by dividing the smoothed GM images by intracranial volume to control for differences in the head size. Then, proportional GM images were converted to GM-BHQ [49], which is similar to the intelligence quotient. The mean value was defined as BHQ 100, and the standard deviation (SD) was defined as 15 BHQ points. Approximately 68% of the population was between BHQ 85 and BHQ 115, and 95% of the population was between BHQ 70 and BHQ 130. The GM-BHQ was calculated based on the database of Nemoto et al. (2017) [49], which contains data from 100 healthy participants (64 women, 80 men; mean age = 48.4 years, SD = 8.1 years). First, the regional GM quotients of 116 brain regions defined by the automated anatomical labelling (AAL) atlas [50] were extracted. Second, 116 regional GM quotients were averaged to produce the participant-specific GM-BHQ according to the database of Nemoto et al. (2017). The graphical flowchart of how GM-BHQ is calculated was shown in a previous study [51].

### 2.10. Statistical Analyses

All statistical analyses were performed using the IBM Statistical Package for Social Sciences Statistics version 26 (IBM, Armonk, NY, USA). Statistical significance was set at *p* < 0.05.

One-way analysis of variance was used to compare age, GM-BHQ before the intervention, and psychological characteristics. We checked variations among the three groups. Gender effects were analysed using a chi-squared test.

GM-BHQ changes were calculated by subtracting the number before the intervention from the number 2 months after the intervention. In short, a positive value indicates an increase in the GM-BHQ after 2 months. A one-tailed t-test was used to examine whether each of the two intervention groups showed greater GM-BHQ changes than the control group. A one-tailed *t*-test was applied because the two interventions can increase GM-BHQ as mentioned in the introduction. As a post hoc analysis, we used one-tailed t-tests to examine changes in the local BHQ of the olfactory cortex, insula, and orbitofrontal cortex, which are thought to be altered by the olfactory training [52,53,54], between the olfactory training and control groups. In addition, the Spearman’s correlation (one-tailed) was used to examine the relationship among GM-BHQ change and psychological characteristics. The training questionnaires and number of correct answers in the olfactory training, and the scores of behaviour change were analysed using a one-tailed t-test to examine whether the learning of neuroplasticity and brain healthcare group showed higher scores than the control and olfactory training groups.

## 3. Results

### 3.1. Demographic Data

One participant in the olfactory training group was excluded because they did not undergo MRI examinations. There was no significant difference among the three groups in age, sex, psychological characteristics, and GM-BHQ scores before the intervention (*p* > 0.05) (Table 1).

### 3.2. Changes in the Brain over 2 Months

There was no significant difference between the groups in terms of days between the first and second MRI intervals (F = 1.6, *p* = 0.21, degree of freedom = 89); mean ± SD: 63.0 ± 2.7 was for the control group, 61.7 ± 3.7 for the olfactory training group, 62.0 ± 2.9 for the learning of neuroplasticity and brain healthcare group.

The GM-BHQ changes over 2 months were as follows: mean ± SD: −0.68 ± 1.39 for the control group, −0.19 ± 1.35 for the olfactory training group, −0.08 ± 1.28 for the learning of neuroplasticity and brain healthcare group. Figure 3 shows the scatter plot of the GM-BHQ changes over 2 months. Appendix A shows brain changes in 116 regions of the AAL atlas. The learning of neuroplasticity and brain healthcare group showed a significantly smaller GM-BHQ loss than the control group (*p* = 0.046, t = 1.72). There was no significant difference between the olfactory training and control groups (*p* = 0.09, t = 1.36). The results of the post hoc analysis, which examined the brain regions that are thought to be altered by the olfactory training, are summarized in Table 4. The olfactory training group showed a significantly smaller local BHQ loss than the control group in the right insula (*p* = 0.02, t =2.07), left orbital middle frontal gyrus (*p* = 0.04, t = 1.73), and left/right medial orbital superior frontal gyrus (*p* = 0.02 and 0.03, t = 2.05 and 1.87, respectively).

The scatter plot indicates grey matter-brain healthcare quotient (GM-BHQ) changes over 2 months in the three groups.

### 3.3. Brain Changes and Psychological Characteristics

Table 2 shows correlations between the GM-BHQ change over 2 months and scores of psychological characteristics. The Curiosity and Exploration Inventory-II scores were significantly correlated with the GM-BHQ change in the olfactory training and learning of neuroplasticity and brain healthcare groups (r = 0.40, *p* = 0.02 and r = 0.56, *p* < 0.01, respectively). In addition, the Personal Growth Initiative Scale-II score and Short Grit Scale were significantly correlated with the GM-BHQ change in the learning of neuroplasticity and brain healthcare group (r = 0.39, *p* = 0.02, r = 0.35, *p* = 0.03, respectively).

### 3.4. Brain Changes and the Number of Correct Answers in the Olfactory Training Questionnaire

Table 3 shows correlations between GM-BHQ change over 2 months and the scores of the training questionnaire. The GM-BHQ change showed a significant positive correlation with the questions, ‘Did you enjoy the training?’, ‘Have you become interested in scents?’, and ‘Has your sense of smell improved?’.

Regarding the number of correct answers in the entire olfactory training, there was no significant correlation with the GM-BHQ change (r = −0.16, *p* = 0.41) (Appendix A).

### 3.5. Behavioural Changes in Three Groups Two Months after Training

The total scores of behaviour change were as follows: mean ± SD: 5.6 ± 3.3 for the control group, 5.3 ± 3.7 for the olfactory training group, 7.3 ± 3.9 for the learning of neuroplasticity and brain healthcare group, which showed significantly higher scores than the control (*p* = 0.03) and olfactory training (*p* = 0.02) groups. The scores of behavioural change questions were significantly correlated with GM-BHQ change over 2 months in learning neuroplasticity and brain healthcare (r = 0.34, *p* = 0.03) and olfactory training (r = 0.42, *p* = 0.02) groups. There was no significant correlation in the control group (r = −0.10, *p* = 0.30).

## 4. Discussion

This study presents two major results. First, the sense of achievement, interest, and curiosity are important for neuroplasticity due to training. Second, learning about neuroplasticity and brain healthcare leads to improved brain health.

### 4.1. Olfactory Training

A decline in the sense of smell may substantially affect quality of life [17]. A recent study revealed that altered smell and taste related with coronavirus disease-2019 can result in severe disruptions to daily living including altered eating, appetite loss, weight change, and relationship to self and others [55]. In addition, olfactory disfunction may be related to depression [56] and neurodegenerative diseases, such as Alzheimer’s and Parkinson’s disease [57]. Therefore, the olfactory training is expected to have various effects [58]. For instance, in older people, the olfactory training was found to improve semantic verbal fluency and decrease the severity of depressive symptoms in individuals with subclinical depression [16]. 

The link between brain anatomy and olfactory function has been investigated in previous studies, which have shown a correlation between olfactory function and the volume of the olfactory bulb [52,53] or cortical thickness of olfactory processing areas, such as the orbitofrontal cortex and insula [53,54]. Perfumers and sommeliers who specialise in olfactory function also showed increased GM volume in the orbitofrontal cortex, insula, and entorhinal cortex compared to healthy controls [59,60]. Further, the olfactory training for healthy participants increased the olfactory bulb volume [20] and cortical thickness [21]. In the current study, the olfactory training group also showed significant differences in GM alterations at 2 months in the insula and orbitofrontal cortex compared to the control group. Contrarily, the olfactory training group did not show a significant difference from the control group in GM-BHQ changes. A recent longitudinal study of sommelier students showed increased cortical thickness in the right entorhinal cortex and decreased cortical thickness in the left inferior temporal gyrus, superior parietal gyrus, superior frontal gyrus, and right pars triangularis due to sommelier training [61]. In the current study, it is possible that mixed increased and decreased GM volume in different brain regions resulted in an unclear change in GM-BHQ, which is an index of GM volume averaged over AAL 116 brain regions. Thus, evaluation in individual brain regions is considered to be required to assess the usefulness of specific training, such as olfactory training.

In contrast, high scores on the sense of achievement and interest exams were positively associated with GM-BHQ changes. This result suggests that a sense of achievement and interest towards training is important for neuroplasticity. Furthermore, subjective improvement in the sense of smell was associated with increased GM-BHQ scores. In contrast, the number of correct answers in the entire olfactory training was not significantly correlated with GM-BHQ changes. Interestingly, our results suggest that the attitude towards training and growth through training is more important than obtaining a high score. In addition, while a variety of training and learning activities can alter the brain, it is effective for the brain reserve to choose tasks that are of interest.

### 4.2. Learning Neuroplasticity and Brain Healthcare

The learning of neuroplasticity and brain healthcare group showed a significantly lower GM-BHQ decline after the 2-month interval and a higher mean score for behavioural change. This result suggests that learning neuroplasticity and brain healthcare can alter behaviour and lead to good brain health. This result is also consistent with previous studies; neuroplasticity knowledge can alter behaviour [23], and health messages can lead to healthy behaviour via brain activation [24,25]. Learning about neuroplasticity and brain healthcare is one way to improve the function of the brain reserve.

However, it is still unknown whether GM changes due to temporary intervention, as in this study, have protective effects against long-term decline of GM volume and brain ageing. Only a few studies have reported a even a short-term follow-up after the intervention. One study reported a reasonable course of GM changes after juggling training. After a 3-month training, participants acquired juggling skills and showed increased GM volume in the mid-temporal area and posterior intraparietal sulcus. After an additional 3 months without juggling training, the participants lost their juggling skills, and GM volume decreased [62]. In contrast, a complex course of GM change was reported by a study that investigated three different time points at which medical students prepared for their medical examination [9]. The GM increased in the posterior and lateral parietal cortices during the learning period. These structural changes did not change significantly during the semester break 3 months after the exam. Interestingly, the posterior hippocampus showed continuous expansion during the learning period and semester breaks. Thus, the course of GM change after training could differ depending on the training content and brain region. To the best of our knowledge, no MRI study has investigated the long-term course after the end of a temporary intervention. However, the Advanced Cognitive Training for Independent and Vital Elderly Cognitive Training study reported that 6-week cognitive training of reasoning and speed-of-processing interventions maintained its effects on the targeted cognitive abilities at 10 years [63]. Long-term MRI studies are needed to link brain reserves and intervention studies with neuroplasticity.

### 4.3. Psychological Characteristics and the Effect of the Intervention

In the current study, the change in GM-BHQ was correlated with curiosity in both the interventions—olfactory training and the learning of neuroplasticity and brain healthcare—whereas the Personal Growth Initiative (PGI) and Short Grit Scale were correlated with the learning of neuroplasticity and brain healthcare group. Curiosity is defined as a positive emotional-motivational system associated with the recognition, pursuit, and self-regulation of novel and challenging opportunities [64]. PGI is a developed set of skills for self-improvement that includes cognition and behaviour [30]. Grit is defined as perseverance and passion for long-term goals [65]. Our results suggest the importance of curiosity, PGI, and Grit for neuroplasticity due to training and learning. Interestingly, in the control group, there was no correlation between GM-BHQ change and curiosity or personal growth initiative. For participants with high curiosity, PGI, and Grit, having some opportunity for interventions may be considered important for the brain reserve.

### 4.4. Length of the Training

In the current study, we employed different lengths of training for two interventions: learning neuroplasticity and brain healthcare messages for 28 days and olfactory training for 40 days. The length of the training and time before the second MRI could influence the results of this study. A previous study suggested that GM alternations captured by MRI occur very quickly after the start of training. Kwok et al. reported that learning newly defined and named subcategories in a period of 2 h increased GM volume of the visual cortex [66]. Interestingly, this increased GM volume can renormalise. After 3 months of juggling training, GM volume in the midtemporal area increased, but the volume in the same area decreased 3 months after training stopped [62]. Further, increased GM volume can also be renormalised during periods of ongoing training. In a previous study that measured GM alternations during 7 weeks of left-handed writing and drawing, GM volume of the primary motor cortex increased during the first 4 weeks and then partially renormalised despite continued training [67]. In contrast, long-term training can increase GM volume in the corresponding brain areas. For example, professional musicians exhibit larger GM volumes in the auditory and motor cortices than non-musicians [68]. However, the appropriate type and length of training are still not well understood from the perspective of preventing age-related GM decline and brain reserve. It is particularly desirable to elucidate whether temporary training-induced GM alterations can prevent age-related GM decline in the long term.

### 4.5. Limitations

This study had some limitations. First, the sample size was small, which reduced the power of statistical analyses. Second, a randomised crossover trial was not conducted. For instance, the control group showed a mean GM-BHQ change of −0.68 over 2 months. Because the mean GM-BHQ deviation value was defined as 100, the mean GM-BHQ changes in the control group were higher than expected. In a previous study [69], the annual GM volume decline rate was less than 0.4% at 50 years of age, which was the average age of the participants in this study. Although we conducted a controlled study with two intervention training groups and a control group, additional studies are desirable. Third, the functional changes in the sense of smell were evaluated using a subjective questionnaire. Fourth, we did not perform olfactory testing. Some participants showed a low number of correct answers in the olfactory training group (Appendix A), indicating the possibility of individuals with olfactory dysfunction being included. Fifth, two types of interventions were not directly contrasted in the analysis, because, at the study design stage, we did not set the conditions to compare the two interventions (i.e., 28-day messages for learning neuroplasticity and brain healthcare vs. 40-day training sessions for the olfactory training). Sixth, we did not correct for multiple comparisons, although we performed planned comparisons.

## 5. Conclusions

Due to learning and training, attitudes towards training, such as the sense of achievement, interest, and psychological characteristics of high curiosity, are important for neuroplasticity. When starting training and learning for the betterment of the brain reserve, employing initiatives that interest you may be effective. In addition, learning neuroplasticity and brain healthcare can alter behaviour and improve brain health.

## Figures and Tables

**Figure 1 life-13-00667-f001:**
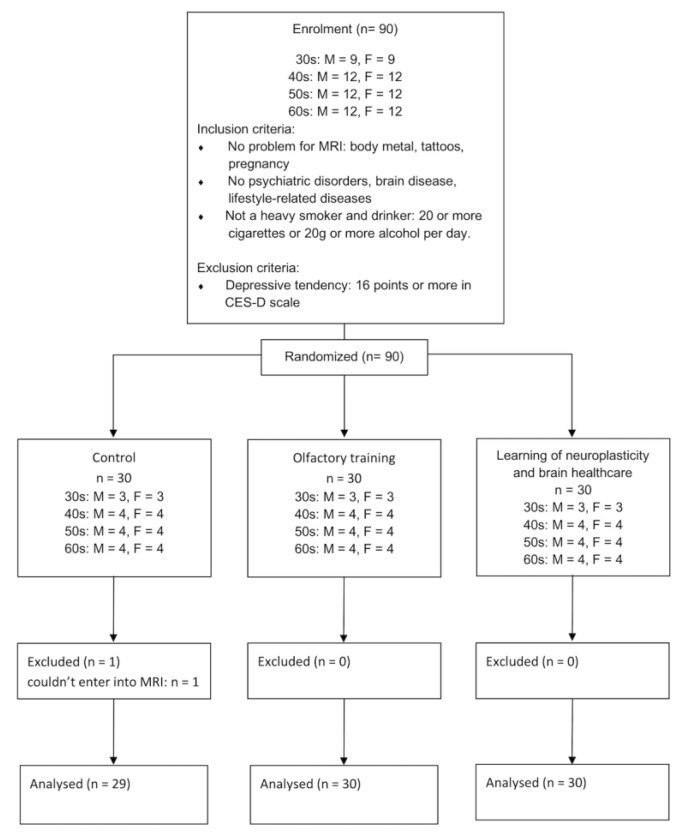
Flow chart.

**Figure 2 life-13-00667-f002:**
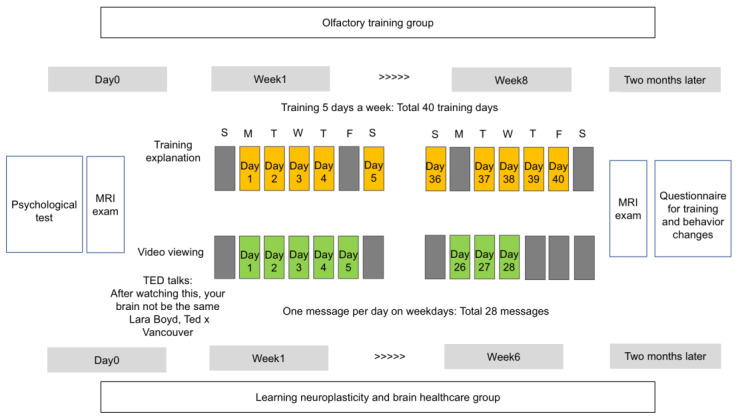
Experimental design of the study and interventions.

**Figure 3 life-13-00667-f003:**
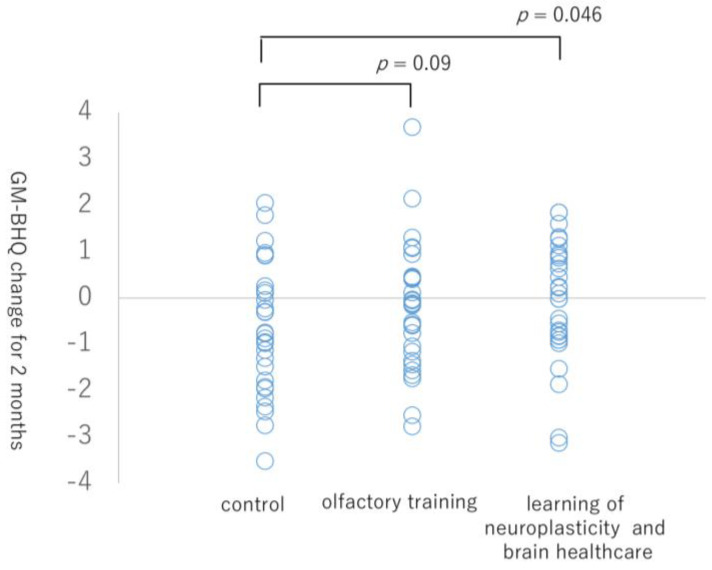
Scatter plot of grey matter changes.

**Table 1 life-13-00667-t001:** Demographic data for grey matter-brain healthcare quotient (GM-BHQ). SD, standard deviation.

	Control	Olfactory Training	Learning of Neuroplasticity and Brain Healthcare	*F*	*p*
Age, mean (range, SD)	49.9 (46.0–53.8, 10.2)	49.8 (46.0–53.5, 10.0)	50.7 (46.8–54.6, 10.4)	0.07	0.93
Number of females	14	15	15	-	0.99
GM-BHQ, mean (range, SD), mean (range, SD)	101.1 (84.0–113.4, 7.3)	99.2 (84.9–108.5, 6.3)	100.4 (89.0-116.3, 6.6)	0.63	0.53
Personal Growth Initiative Scale-II, mean (range, SD)	58.2 (19–83, 15.7)	54.6 (21–79, 14.6)	50.2 (16–96, 18.1)	1.77	0.18
Satisfaction with life scale, mean (range, SD)	17.5 (5–32, 7.4)	14.7 (5–33, 7.4)	17.9 (5–30, 7.2)	1.66	0.20
Subjective Happiness Scale, mean (range, SD)	16.6 (10–21, 2.5)	16.0 (10–26, 3.8)	17.1 (11–26, 3.3)	0.82	0.45
The Curiosity and Exploration Inventory-II, mean (range, SD)	30.1 (16–44, 7.5)	31.2 (19–48, 6.0)	28.2 (10–41, 8.7)	1.21	0.30
Short Grit Scale, mean (range, SD)	27.6 (10–38, 6.4)	25.5 (16–32, 4.1)	26.3 (12–35, 5.6)	1.15	0.32
Rosenberg Self Esteem Scale, mean (range, SD)	27.3 (15–39, 5.6)	23.8 (12–39, 6.7)	26.5 (15–38, 6.3)	2.63	0.08
Perceived Stress Scale, mean (range, SD)	37.2 (21–52, 6.4)	40.1 (30–52, 5.4)	38.1 (26–50, 6.4)	1.69	0.19

**Table 2 life-13-00667-t002:** Correlation between psychological characteristics and grey matter-brain healthcare quotient change.

	Control		Olfactory Training		Learning of Neuroplasticity and Brain Healthcare	
	r	*p*	r	*p*	r	*p*
Personal Growth Initiative Scale-II	−0.09	0.32	0.29	0.07	0.39 *	0.02
Satisfaction with life scale	0.19	0.16	0.08	0.35	0.18	0.18
Subjective Happiness Scale	0.22	0.13	0.24	0.11	0.24	0.10
The Curiosity and Exploration Inventory-II	0.27	0.8	0.40 *	0.02	0.56 *	<0.01
Short Grit Scale	0.19	0.17	0.29	0.07	0.35 *	0.03
Rosenberg Self Esteem Scale	−0.08	0.34	−0.04	0.43	0.17	0.18
Perceived Stress Scale	0.22	0.13	0.12	0.26	−0.18	0.17

* Asterisks mean significance of *p* < 0.05.

**Table 3 life-13-00667-t003:** Correlation between the training questionnaire and grey matter-brain healthcare quotient change in the olfactory training group.

	r	*p*
Did you enjoy the training?	0.34	0.07
Did you feel a sense of achievement?	0.42 *	0.02
Did you feel a burden from training?	−0.20	0.29
Did you feel fatigued from training?	−0.19	0.31
Have you become interested in scents?	0.36 *	0.048
Has your sense of smell improved?	0.37 *	0.04
Has your sense of taste improved?	0.15	0.44

* Asterisks mean significance of *p* < 0.05.

**Table 4 life-13-00667-t004:** Changes in brain regions related to olfactory function over 2 months.

	Control	Olfactory Training	t	*p*
Left Insula	−0.33 ± 2.00	−1.00 ± 1.80	1.39	0.08
Right Insula	−0.05 ± 2.10	−1.10 ± 1.90	2.07	0.02
Left Olfactory cortex	−0.01 ± 3.34	−0.35 ± 2.34	0.45	0.33
Right Olfactory cortex	0.18 ± 3.00	−0.82 ± 2.60	1.37	0.09
Left orbital Superior frontal gyrus	0.07 ± 3.20	−1.00 ± 2.80	1.38	0.09
Right orbital Superior frontal gyrus	−0.02 ± 2.56	−0.68 ± 2.30	1.02	0.16
Left orbital Middle frontal gyrus	0.06 ± 3.40	−1.40 ± 3.10	1.73	0.04
Right orbital Middle frontal gyrus	−0.32 ± 4.60	−0.70 ± 2.50	0.39	0.35
Left medial orbital Superior frontal gyrus	−0.10 ± 2.90	−1.60 ± 3.00	2.05	0.02
Right medial orbital Superior frontal gyrus	−0.18 ± 2.80	−1.40 ± 2.30	1.87	0.03
Left orbital Inferior frontal gyrus	−0.50 ± 2.60	−1.20 ± 2.70	0.99	0.16
Right orbital Inferior frontal gyrus	−0.74 ± 2.40	−0.97 ± 1.90	0.4	0.35

## Data Availability

Not applicable.

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
