# Peer review of "Altered Grey Matter-Brain Healthcare Quotient: Interventions of Olfactory Training and Learning of Neuroplasticity"

_life, 2023, doi:10.3390/life13030667_

Round 1
Reviewer 1 Report
Reviewer’s comment
This study measured the Gray matter brain healthcare quotient (GM-BRQ) as a surrogate measure of GM volume changes in response to the two interventions and compared the quotient to the control group before the interventions. Sense of interest, curiosity, and interest was critical in neuroplasty. Learning neuroplasty increases the brain reserve for neuroplasty. The topic is interesting and shows potential in terms of preventing the decline of aging with learning neuroplasty itself. However, some issues should be clarified clearly before publication.
Major comments
1. Why do the numbers of training days differ between the olfactory and learning neuroplasty groups? It appears that the learning neuroplasty group has more time until the second MRI exam. The difference in time before the second MRI after the final training session may affect the results?
2. For learning neuroplasty group. It seems that participants in this group watched Youtube first, and after that, healthcare messages were emailed to them each 28 days. How was the compliance of watching this video assessed in this study? Did they watch Youtube at home or experimental place? Are there participants easily distracted or not focused while watching youtube, although a brief test was done to check whether they understood the content of the TED youtube? Any differences in GM- BRQ between well-focused participants and the others?
3. Is there any other reason not to compare GM-BRQ between the two groups after the training? In this study, the authors compared GM -BRQ for the two groups with only the control group without comparing with each other, although this was commented on the limitation in the discussion part.
4. Questionnaire to evaluate behavioral changes: It remains unclear how behavioral changes were assessed with the questionnaire for three groups. Was the questionnaire the same for all groups? Regarding Table 3, the questions in table 3 are specific to the only olfactory group? If so, what is the meaning of the total scores described in the 3.5 section? If the questionnaire is different among the three groups, how can we interpret the difference in total numbers reflecting behavioral changes among the groups? It is really confusing.
5. Is there any reason to use a one-tailed t-test, not a two-tailed t-test, to compare GM-BHQ changes of the two intervention groups with the control group?
6. The authors speculate that for the olfactory training group, the difference in GM-BHQ between the learning neuroplasty group may be rooted in the training-related decrease in other brain areas, leading to overall GM-BRQ reductions. Please describe why olfactory training enhances the decline in GM-BHQ in more detail. Do we understand that, generally, when the brain is activated on a specific function, the other brain areas are inactivated to increase the contrast that the brain is focused on? In which areas were GM volumes decreased by olfactory training?
Author Response
We would like to thank you for reviewer’s insightful comments, which have greatly helped us to improve the quality of our manuscript.
Reviewer #1
This study measured the Gray matter brain healthcare quotient (GM-BRQ) as a surrogate measure of GM volume changes in response to the two interventions and compared the quotient to the control group before the interventions. Sense of interest, curiosity, and interest was critical in neuroplasty. Learning neuroplasty increases the brain reserve for neuroplasty. The topic is interesting and shows potential in terms of preventing the decline of aging with learning neuroplasty itself. However, some issues should be clarified clearly before publication.
Major comments
- Why do the numbers of training days differ between the olfactory and learning neuroplasty groups? It appears that the learning neuroplasty group has more time until the second MRI exam. The difference in time before the second MRI after the final training session may affect the results?
Our reply:
The numbers of training days differ because we did not consider a comparison between the olfactory and learning neuroplasty groups. Therefore, we mentioned this point as limitation in 4.5. Limitation section as follows; “Because, at the study design stage, we did not set the conditions to compare the two interventions (i.e. 445 28-day messages for learning neuroplasticity and brain healthcare vs 40-day training sessions for the olfactory training).”.
We also consider that the difference in time before the second MRI can affect the results. We revised the sentence in 4.4 Length of the training to “The length of the training and time before the second MRI could influence the results of this study.”.
- For learning neuroplasty group. It seems that participants in this group watched Youtube first, and after that, healthcare messages were emailed to them each 28 days. How was the compliance of watching this video assessed in this study? Did they watch Youtube at home or experimental place? Are there participants easily distracted or not focused while watching youtube, although a brief test was done to check whether they understood the content of the TED youtube? Any differences in GM- BRQ between well-focused participants and the others?
Our reply:
The participants watch TED YouTube at experimental space. Further, all participants either got a perfect score or got one question wrong.] As you suggested, there is difference between well-focused participants and the others. However, in this study, we expected that all participants focused on TED YouTube.
We revised the sentence in 2.4. Learning neuroplasticity and brain healthcare to “In the session on learning neuroplasticity at experimental space, the participants watched a YouTube video of the TED Talk by Lara Boyd on the same topic”.
In addition, we added the following sentence in 2.4. Learning neuroplasticity and brain healthcare; “All participants either got a perfect score or got one question wrong.”.
- Is there any other reason not to compare GM-BRQ between the two groups after the training? In this study, the authors compared GM -BRQ for the two groups with only the control group without comparing with each other, although this was commented on the limitation in the discussion part.
Our reply:
In this study, we did not design the study to compare the superiority of the olfactory training and learning neuroplasticity for brain healthcare. Therefore, we considered that a comparison between two intervention groups can be a nuisance analysis.
- Questionnaire to evaluate behavioral changes: It remains unclear how behavioral changes were assessed with the questionnaire for three groups. Was the questionnaire the same for all groups? Regarding Table 3, the questions in table 3 are specific to the only olfactory group? If so, what is the meaning of the total scores described in the 3.5 section? If the questionnaire is different among the three groups, how can we interpret the difference in total numbers reflecting behavioral changes among the groups? It is really confusing.
Our reply:
We apologize my mistake. I forgot to attach Supplemental file of Questionnaire to evaluate behavioral changes. I added the Supplemental file including questionnaire to evaluate behavioral changes.
All groups answered the Questionnaire to evaluate behavioral changes. In addition, the questions in table 3 are specific to the only olfactory group. To state clearly, we revised the sentences in 2.6. Questionnaire for olfactory training and 2.7. Questionnaire for behavioural changes as follows; “A questionnaire for only olfactory training was administered after the second MRI acquisition to investigate attitudes towards training.” and “Therefore, we asked all participants of the three groups using a questionnaire to determine whether they started a new initiative between the first and second MRI.”
- Is there any reason to use a one-tailed t-test, not a two-tailed t-test, to compare GM-BHQ changes of the two intervention groups with the control group?
Our reply:
According to the previous studies, the trainings such as the olfactory training can increase GM volume as whole brain. Therefore, we considered that to test the statistical significance in one direction is an appropriate statistical analysis method.
We added following sentence in 2.10. Statistical analyses; “A one-tailed t-test was applied because the two intervention can increase GM-BHQ as mentioned in introduction part.”.
- The authors speculate that for the olfactory training group, the difference in GM-BHQ between the learning neuroplasty group may be rooted in the training-related decrease in other brain areas, leading to overall GM-BHQ reductions. Please describe why olfactory training enhances the decline in GM-BHQ in more detail. Do we understand that, generally, when the brain is activated on a specific function, the other brain areas are inactivated to increase the contrast that the brain is focused on? In which areas were GM volumes decreased by olfactory training?
Our reply:
We appreciate your comment.
In the current study, the olfactory training group showed smaller GM-BHQ loss over two months compared to the control group, although it was not statistically significant (p = 0.09, Figure 3). In addition, GM-BHQ changes was significantly correlated with the questionnaire of “Did you feel a sense of achievement?”, which suggests that participants with a sense of achievement increase GM-BHQ.
Therefore, we consider that the olfactory training can lead to overall GM-BHQ increases or suppress decrease in GM volume or GM-BHQ due to age-related changes or others. It is not our intention that the olfactory training can lead to overall GM-BHQ.
To state the brain areas where GM volumes decreased by olfactory training, we revised a sentence in 4.1. Olfactory training to “A recent longitudinal study of sommelier students showed increased cortical thickness in the right entorhinal cortex and decreased cortical thickness in the left inferior temporal gyrus, superior parietal gyrus, superior frontal gyrus, and right pars triangularis due to sommelier training” from “A recent longitudinal study of sommelier students showed increased cortical thickness in the right entorhinal cortex and decreased cortical thickness in other cerebral regions due to sommelier training”.
Reviewer 2 Report
From the Abstract, it is not clear to me which is the state-of-the-art and the related rationale for this study.
In the Introduction, as well as in the Discussion, more insight about the usefulness of olfactory training in the improvement of the overall quality of life of individuals is desired.
Overall, I would recommend a lookout on future studies stemming from this and with respect to other literature evidences about olfaction (e.g., to use training to improve emotions, to treat behavioral or food-related disturbances, and so forth, maybe using more cost-affordable technologies than MRI is). Please, also refer to more recent literature, which is quite important in the recent years in the specific domain.
Some typos are present throughout the manuscript. A thorough proofread of the work can be helpful.
Author Response
We would like to thank you for reviewer’s insightful comments, which have greatly helped us to improve the quality of our manuscript.
Reviewer #2
- From the Abstract, it is not clear to me which is the state-of-the-art and the related rationale for this study.
Our reply:
According to your comment, we revised the abstract as follows;
Recent studies revealed that grey matter (GM) changes due to various training and learning using magnetic resonance imaging. In this study, we investigate the effect of psychological characteristics and attitudes toward training and learning on GM changes. Ninety participants were recruited and distributed into three groups: an olfactory training group that underwent 40 olfactory training sessions designed for odour classification tasks, a group classified for learning of neuroplasticity and brain healthcare using a TED Talk video and 28 daily brain healthcare messages, and a control group. Further, we assessed psychological characteristics, such as curiosity and personal growth initiative. In the olfactory training group, we conducted a questionnaire survey on olfactory training regarding their interests and sense of accomplishment. In the olfactory training group, the GM change was significantly correlated with the sense of achievement and interest in training. The learning of neuroplasticity and brain healthcare group showed a significantly smaller 2-month GM decline than did the control group. The Curiosity and Exploration Inventory-II scores were significantly correlated with GM changes in both intervention groups only. In conclusion, our result suggested that training or learning with a sense of accomplishment, interest, and curiosity would lead to greater GM changes.
- In the Introduction, as well as in the Discussion, more insight about the usefulness of olfactory training in the improvement of the overall quality of life of individuals is desired.
Overall, I would recommend a lookout on future studies stemming from this and with respect to other literature evidences about olfaction (e.g., to use training to improve emotions, to treat behavioral or food-related disturbances, and so forth, maybe using more cost-affordable technologies than MRI is). Please, also refer to more recent literature, which is quite important in the recent years in the specific domain.
Our reply:
We appreciate your suggestion. We added a following sentence in Introduction; “ In addition, it has been suggested that olfactory training not only improves the sense of smell, but also may lead to improved quality of life.”.
We also added a following section in 4.1. Olfactory training;
A decline in the sense of smell may substantially affect quality of life [17]. A recent study revealed that altered smell and taste related with coronavirus disease-2019 can result in severe disruptions to daily living including altered eating, appetite loss, weight change, and relationship to self and others [55]. In addition, olfactory disfunction may be related to depression [56] and neurodegenerative diseases, such as Alzheimer’s and Parkinson’s disease [57]. Therefore, the olfactory training is expected to have various effects [58]. For instance, in older people, the olfactory training was found to improve sematic verbal fluency and decrease the severity of depressive symptom in individuals with subclinical depression [16].
- Some typos are present throughout the manuscript. A thorough proofread of the work can be helpful.
Our reply:
We are sorry about typos. English proofread service had been used before submission but we used English proofread service again (Please see attached file of a certificate ).
Round 2
Reviewer 1 Report
I appreciate the authors' responses to the reviewer’s comments and suggestions, and I am pleased to see that the authors have revised the manuscript accordingly. Thank you for all your work.